# Post-Injury Buprenorphine Administration Is Associated with Long-Term Region-Specific Glial Alterations in Rats

**DOI:** 10.3390/pharmaceutics14102068

**Published:** 2022-09-28

**Authors:** Jane Ryu, Pantea Jeizan, Saira Ahmed, Sareena Ehsan, Jefin Jose, Sean Regan, Karen Gorse, Corrina Kelliher, Audrey Lafrenaye

**Affiliations:** Department of Anatomy and Neurobiology, Virginia Commonwealth University, Richmond, VA 23298, USA

**Keywords:** traumatic brain injury, buprenorphine, Bup-SR-Lab, microglia, astrocyte, myelin, membrane disruption, somatosensory sensitivity

## Abstract

Traumatic brain injury (TBI) is a major leading cause of death and disability. While previous studies regarding focal pathologies following TBI have been done, there is a lack of information concerning the role of analgesics and their influences on injury pathology. Buprenorphine (Bup), an opioid analgesic, is a commonly used analgesic in experimental TBI models. Our previous studies investigated the acute effects of Buprenorphine-sustained release-Lab (Bup-SR-Lab) on diffuse neuronal/glial pathology, neuroinflammation, cell damage, and systemic physiology. The current study investigated the longer-term chronic outcomes of Bup-SR-Lab treatment at 4 weeks following TBI utilizing a central fluid percussion injury (cFPI) model in adult male rats. Histological assessments of physiological changes, neuronal damage, cortical and thalamic cytokine expression, microglial and astrocyte morphological changes, and myelin alterations were done, as we had done in our acute study. In the current study the Whisker Nuisance Task (WNT) was also performed pre- and 4w post-injury to assess changes in somatosensory sensitivity following saline or Bup-SR-Lab treatment. Bup-SR-Lab treatment had no impact on overall physiology or neuronal damage at 4w post-injury regardless of region or injury, nor did it have any significant effects on somatosensory sensitivity. However, greater IL-4 cytokine expression with Bup-SR-Lab treatment was observed compared to saline treated animals. Microglia and astrocytes also demonstrated region-specific morphological alterations associated with Bup-SR-Lab treatment, in which cortical microglia and thalamic astrocytes were particularly vulnerable to Bup-mediated changes. There were discernable injury-specific and region-specific differences regarding myelin integrity and changes in specific myelin basic protein (MBP) isoform expression following Bup-SR-Lab treatment. This study indicates that use of Bup-SR-Lab could impact TBI-induced glial alterations in a region-specific manner 4w following diffuse brain injury.

## 1. Introduction

Traumatic brain injury (TBI) is a major cause of death and disability, precipitating almost 200 deaths each day in 2019 and nearly 2 million reported cases occurring annually in the US alone [1,2,3,4] while in other countries, millions suffer TBIs each year [5,6,7,8]. Brain injuries account for significant healthcare cost and adverse health effects that can persist years following the initial injury. A TBI involves a primary and secondary injury. Primary injuries occur from a mechanical head injury that results as an immediate effect from impact of the initial trauma [9]. Primary injuries can further be defined in terms of distribution of structural damage: focal injury or diffuse injury. Focal injuries of TBI result in a more localized, specific part of the brain and produce homogenous cellular damage. Diffuse brain injuries, on the other hand, involve a more widespread area and often involve widespread alterations or damage to the brain that occurs in pockets of injury and results in heterogeneous cellular responses, which range from diffuse neuronal damage, such as neuronal membrane disruption, to inflammatory changes and myelin pathology. TBI-induced morbidities, including problems with cognition, behavior, and sensory processing, have been shown to be associated with these diffuse pathologies [10,11], however, knowledge regarding diffuse pathologies is limited. Because diffuse pathologies are widely distributed throughout the brain and require single-cell resolution to identify, it is difficult to detect in the human population. Thus, experimental studies utilize animal models for assessments of these TBI-induced diffuse pathologies. Due to the utilization of animal models, the National Research Council and National Institutes of Health [12] recommend the use of analgesics, with the most common being opioids.

Buprenorphine (Bup) is a semi-synthetic opioid clinically used for pain management and opioid addiction. Distinct from other opioids, like morphine, Bup is a partial agonist of the mu-opioid receptor (MOR) and an antagonist at the kappa- (KOR) and delta-receptors (DOR) [13,14,15]. As a partial MOR agonist, Bup binds and activates the receptor but does not elicit the maximum possible response produced by full agonists like morphine. Due to this, Bup has a wider safety margin than full agonist opioids like morphine, meaning there is a lower risk for overdose and more moderate symptoms of withdrawal. More recently, following a traumatic brain injury, partial opiate agonists such as buprenorphine have been used for pain treatment for more severe pain [16]. Buprenorphine’s slow dissociation from the μ-opioid receptor paired with its sustained release formulation (Bup-SR-Lab) allows for prolongation of its analgesic effects over days following a single administration and is thus commonly used in experimental studies [17,18,19,20,21]. However, the potential impact of post-injury acute Bup-SR-Lab exposure on pathological progression and longer-term outcomes following TBI remain unknown as the full effects of opioids on various pathologies have not been widely studied.

Our previous study analyzing the effects of Bup-SR-Lab on acute pathology 1d post-injury found that while acute neuronal pathology was not altered by Bup-SR-Lab administration, significant region and drug related changes in glial responses were observed [22]. The effects of Bup-SR-Lab on cellular and neuronal pathologies, inflammation, and somatosensory sensitivity following injury, however, are still unclear. Therefore, the current study focused on evaluating the chronic effects of Bup-SR-lab following brain injury.

## 2. Materials and Methods

### 2.1. Animals

Experiments were conducted under the approve number AM10251 in accordance with ARRIVE guidelines [23] and the Virginia Commonwealth University institutional ethical guidelines concerning the care and use of laboratory animals and were approved by the Institutional Animal Care and Use Committee at Virginia Commonwealth University, which adhere to regulations including, but not limited to, those set forth in the “Guide for the Care and Use of Laboratory Animals: 8th Edition” (National Research Council [12]). Overall, 24 adult (12–16-week old) male Sprague Dawley rats were used in this study. A random number generator determined the animal group (n = 6/group) prior to surgery (1) Sham injury with saline treatment, (2) Sham injury with Bup-SR-Lab treatment, (3) TBI with Saline treatment, or (4) TBI with Bup-SR-Lab treatment. Animals were housed in individual cages on a 12 h light-dark cycle with free access to food and water and full veterinary oversight.

### 2.2. Surgical Preparation, Injury Induction, and Drug Administration

Surgeries were done as previously described [22]. Specifically, anesthesia was induced with 4% isoflurane in 30% O_2_/70% room air. Animals were then ventilated with 1.5–2.5% isoflurane in 30% O_2_ and 70% room air throughout the duration of the surgery, injury, and physiologic monitoring. Body temperature was maintained at 37 °C with a rectal thermometer connected to a feedback-controlled heating pad (Harvard Apparatus, Holliston, MA, USA). All animals were placed in a stereotaxic frame (David Kopf Instruments, Tujunga, CA, USA). A midline incision was made followed by a 4.8 mm diameter circular craniectomy, which was positioned along the sagittal suture midway between bregma and lambda. The dura was left intact. Bone wax was used to seal the burr hole used for the ICP measurements before preparation for central fluid percussion injury (cFPI). The procedures used to induce cFPI were consistent with those described previously [22,24,25,26]. Briefly, a Luer-Loc syringe hub was affixed to the craniotomy site with dental acrylic (methyl methacrylate; Hygenic, Akron, OH, USA) that was applied around the hub, including the area overlying the sealed burr hole and allowed to harden. Animals were removed from the stereotaxic frame and placed on a raised platform for connection to the fluid percussion device, maintaining an unbroken fluid-filled system from the intact dura through the cylinder, via a Luer-Loc adaptor. During injury the investigator supported the animal’s body on the platform but did not hold the head allowing the Luer-Loc mechanism to maintain connection between the injury hub and fluid percussion device. To induce a mild-moderate cFPI a pendulum was released onto the fluid-filled cylinder of the FPI device, producing a pressure pulse of 2.05 ± 0.10 atmospheres for ~22.5 ms (Table 1), which was transduced through the intact dura to the CSF. The pressure pulse was measured by a transducer affixed to the injury device and displayed on an oscilloscope (Tektronix, Beaverton, OR, USA). For sham-injured animals, the animals underwent the same procedure, except the pendulum was not released. Immediately following the injury, animals were replaced in the stereotaxic device. The hub, dental acrylic, and bone wax were removed en bloc and Gelfoam was placed over the craniectomy site. The ICP probe was reinserted into the lateral ventricle, as described above, for post-injury ICP monitoring. Immediate post-injury physiology was recorded for 15 min after cFPI followed by subcutaneous administration of the veterinarian and pharmaceutical company recommended dose of 1 mg/kg Bup SR-Lab or equal volume of saline. The randomized treatment was administered by another investigator to avoid unblinding of the surgeon due to the notable difference in viscosity of the solutions that might introduce bias. One hour following injury the scalp was sutured and treated with lidocaine and triple-antibiotic ointment. Rats were then allowed to recover and were returned to clean home cages. Two hours prior to sacrifice all animals were re-anesthetized for dextran infusion into the left lateral ventricle, as described previously [22,27]. Briefly, The scalp was incised and a 1 mm diameter bur hole was drilled into the skull 1 mm posterior and 0.8 mm lateral to bregma. A spinal needle connected to a microinfusion pump and a pressure transducer was inserted into the left lateral ventricle. Upon breaching the lateral ventricle 15 μL of 40 mg/mL solution of fluorescently tagged 10 kDa dextran (either 488-Alexa Fluor Cat.#: D22910, or 568-Alexa Fluor Cat#: D22912, Invitrogen, Carlsbad, CA, USA) was infused at a rate of 0.5–1.3 μL/min with continuous intracranial pressure monitoring. The dextan was allowed to permeate through the parenchyma of the cortex for 2 h prior to transcardial perfusion.

### 2.3. Physiologic Assessment

Heart rate, respiratory rate, and hemoglobin oxygen saturation were monitored via a hind paw pulse oximetry sensor (STARR Life Sciences, Oakmont, PA, USA) for the duration of anesthesia, except during the induction of injury. Intracranial pressure (ICP) was measured intraventricularly at 4w post-injury. For this, animals were anesthetized 4w post-injury, as described above, and a 2 mm diameter burr hole was drilled into the left parietal bone overlaying the left lateral ventricle (0.8 mm posterior, 1.3 mm lateral, and 2.5 to 3 mm ventral relative to bregma) through which a 25-gauge needle, connected to a pressure transducer and a micro infusion pump 11 Elite syringe pump (Harvard Apparatus) via PE50 tubing, was placed into the left ventricle. Appropriate placement of the infusion pump into the lateral ventricle was verified via a 2.3 μL/min infusion of sterile saline within the closed fluid pressure system during needle placement [22,24,27]. The needle was held in the lateral ventricle for at least 5 min to record ICP. All physiologic measurements were recorded using a PowerLab System (AD Instruments, Colorado Springs, CO, USA). All animals maintained systemic physiological homeostasis throughout the experiment (i.e., heart rate > 200 beats per min and oxygenation > 90%; Table 1; Figure 1A,B). Intercranial pressure was also consistent across groups (Figure 1C). Recovery time following cFPI (the time from withdrawal of inhaled anesthetic to first movement) and weight loss (percent reduction in animal weight from pre-injury to 4w post-injury) were also assessed.

### 2.4. Whisker Nuisance Task

In order to investigate changes in somatosensory sensitivity following Bup-SR-Lab treatment after TBI, the Whisker Nuisance Task was performed by an investigator blind to animal group pre- and 4w post-injury, as has been described previously [24,28,29,30,31]. Briefly, a plastic open-field test arena lined with an absorbent pad was used and animals were acclimated to the testing area for 5 min prior to testing. To perform the whisker nuisance task, the whiskers of the animals were stimulated on both sides with the tip of a wooden applicator stick for three 5-min-long trials with a 1 min period of non-stimulation in between the three stimulation trials. All tests were scored live and digitally video recorded for analysis by three additional investigators blind to animal group. Behaviors observed included (1) movement, (2) stance and body position, (3) breathing, (4) whisker position, (5) whisker response, (6) evading stimulation, (7) response to stick presentation, (8) grooming, (9) ear position, (10) sniffing, and (11) fur ruffling. To assess the response to the sensory stimulation, each behavioral response was scored on a 0 to 2 point scale (0 = absent, 1 = present, 2 = profound). The individual behavior scores for each animal were summed for each trial and the sums were averaged for the three trials with the highest possible score being 22. A higher score indicates more pronounced agitation/sensitivity. The final scores for each investigator were averaged for each animal.

### 2.5. Tissue Processing

At 4w post-cFPI anesthetized rats were overdosed with Euthasol euthanasia-III solution (Henry Schein, Dublin, OH, USA) followed by transcardial perfusion with cold 0.9% saline. As was described previously [22,25,27] and due to the bilateral nature of cFPI, both fresh and fixed brain tissue were collected from each animal. During the cold saline perfusion, a tissue core of the right lateral neocortex and thalamus/midbrain was taken and stored at −80 °C for molecular assessments prior to transcardial fixation of the remaining tissue with 4% paraformaldehyde/0.2% glutaraldehyde in Millonig’s buffer (136 mmol/L sodium phosphate monobasic/109 mmol/L sodium hydroxide) which allowed for immunohistochemical analysis of the left side of the brain. After transcardial perfusion, the left side of the brain was removed and postfixed for >72 h. Postfixed brains were sectioned coronally in 0.1 mmol/L phosphate buffer with a vibratome (Leica, Banockburn, IL, USA) at a thickness of 40 μm from bregma to ∼4.0 mm posterior to bregma. Sections were collected serially in 12-well plates and stored in Millonig’s buffer at 4 °C. All histological quantitative analyses were performed at least 1 mm posterior to the needle track used for ICP monitoring. The well from which sections would be taken for analysis was selected via a random number generator (1–12) and the first 4 sections with visible hippocampus were taken representing serial sections throughout the rostral-caudal extent (1.8 mm ± 0.2 mm to 3.8 mm ± 0.2 mm posterior to bregma, each 480 μm apart). Histologic analyses were performed on the left lateral somatosensory cortex restricted to layers V and VI extending from the area lateral to CA1 to the area lateral to CA3 of the hippocampus and entire left hemi-thalamus extending from the midline and dorsal surface of the thalamus to the reticular nucleus and zona inserta of the thalamus laterally and ventrally.

### 2.6. Assessment of Cell Damage/Loss

To evaluate the numbers of total and damaged cells in the cortex and thalamus of rats following sham injury or cFPI and vehicle or Bup-SR-Lab treatment, four sequential, randomly selected sections per animal were stained with hematoxylin and eosin (H&E) and assessed as described previously [22,24,27]. Briefly, tissue was mounted on gelatin-coated slides before dehydration and rehydration. Rehydrated tissue was incubated in Gills hematoxylin (Leica Biosystems, Buffalo Grove, IL, USA) followed by bluing agent (Leica Biosystems) and three dips in 0.25% eosin Y/0.005% acetic acid/95% ethanol before sections were cleared through increasing concentrations of ethanol and cover-slipped with Permount (Thermo Fisher Scientific, Waltham, MA, USA). Each full region of interest (Left neocortex and Left hemithalamus) for each section was imaged using a Keyence BZ-X800 microscope (Keyence Corporation of America, Itasca, IL, USA). Total cell number and average cell size were analyzed using the Analyze particles plugin in ImageJ FIJI software following a background subtraction with a 50 pixel rolling ball radius to reduce staining variability [32]. The number of damaged neurons, delineated by eosinophilic cytoplasm and condensed nuclei, in the entire left lateral neocortex and thalamus was counted by three independent investigators blinded to the animal group and all investigator counts were averaged for each animal.

### 2.7. Assessment of Neuronal Membrane Disruption

Consistent with previous studies, we assessed the potential for neuronal membrane disruption via the utilization of fluorescently tagged (Alexa-568 or Alexa-488) 10 kDa dextran, which are impermeable to cells with intact membranes [24,27,33]. Cells containing dextran, therefore, indicate membrane disruption. As the dextrans were fluorescently tagged, no immunolabeling was required for visualization of membrane disruption. Animals in which the dextran infusion was incomplete or compromised were excluded from membrane disruption assessments (3 animals for this study; 1 sham saline and 2 TBI saline). Sections were stained for NeuroTrace fluorescent Nissl stain (Cat # N21479, ThermoFisher Scientific, Waltham, MA, USA) to identify neurons. Staining for all tissue was done at the same time to reduce run-to-run variability. Sections were imaged with a Keyence BZ-X800 microscope (Keyence Corporation of America, Itasca, IL, USA). Images of the left neocortex layers V and VI were taken at ×20 magnification in a systematically random fashion by an investigator blinded to animal group using NeuroTrace to verify focus and location within the region of interest. Counts of NeuroTrace+ neurons containing the cell-impermeable dextran were performed by an investigator blinded to animal group using the FIJI ImageJ [32] ROI manager and Cell Counter plug-in. The percent of NeuroTrace+ membrane disrupted neurons was quantified for each image and averaged for each animal. Individual animals were considered ns.

### 2.8. Immunohistochemistry

Immunohistochemistry was done as previously described [22,34]. Fluorescent immunohistochemistry against the calcium binding protein, Iba-1 (microglia), glial fibrillary acidic protein, GFAP (astrocytes), or myelin basic protein, MBP (myelin) was done to identify microglia, astrocytes, and myelin, respectively. Briefly, 40 μm thick coronal sections were blocked and permeabilized in 1.5% triton and 5% normal goat serum followed by overnight incubation with primary antibody rabbit anti-Iba-1 (Cat. #019-19741, 1:1000, Wako; Osaka, Japan), mouse anti-GFAP (Cat.#MAB3402, 1:1000; GA5, Millipore, Burlington, MA, USA) or mouse anti-myelin basic protein (Cat #808401; 1:1000; SMI99, BioLegend, San Diego, CA, USA) followed by incubation with Alexa Fluor 488-conjugated goat anti-rabbit secondary antibody (Cat.# A11034, 1:700; ThermoFisher Scientific) or Alexa Fluor 568-conjugated goat anti-mouse secondary antibody (Cat.# A-11031, 1:700; ThermoFisher Scientific). Tissue was mounted using Vectashield hardset mounting medium with DAPI (Cat.#H-1500; Vector Laboratories). Immunolabeling for all tissue was done at the same time to reduce run-to-run variability. Imaging was done by an investigator blind to animal group on a Keyence BZ-X800 Microscope (Keyence Corporation of America, Itasca, IL, USA) with all image acquisition settings being held consistent for each region of interest (left lateral neocortex layers V and VI or the left hemithalamus). Four tissue sections were imaged for each animal and four 20× magnification images were taken for each section and each region of interest (16 total images/region of interest/animal). DAPI nuclear labeling was used to verify focus and restriction within the regions of interest prior to image acquisition. All data was recorded by an investigator blind to animal group.

### 2.9. Assessment of Microglial Morphology

Microglial morphology was assessed as previously [22,35]. Briefly, images labeled for Iba-1 were processed with background subtraction and automatic thresholding to generate masks of Iba-1+ microglia. All cells within the mask were added to the Region of Interest Manager in FIJI ImageJ [32] and the average size of each microglial cell was assessed. All Iba-1+ microglial cells in each image were then morphologically analyzed for process number, number of end/terminal points, and maximum process segment length using the Skeletonize and AnalyzeSkeleton tools in FIJI ImageJ [32]. A complexity index was also calculated for each microglia using the formula, complexity index = number of processes/number of end points, with a lower number indicating reduced process complexity. For morphological metrics individual microglia were considered ns.

### 2.10. Assessment of Astrocyte Morphology

Astrocyte morphology was assessed as we have done previously [22]. To make masks of GFAP+ astrocytes, images labeled for GFAP were background subtracted and made binary using FIJI/ImageJ [32]. All cells within the mask were added to the Region of Interest Manager in FIJI/ImageJ [32] and measurements of number of astrocytes/image, cellular area and circularity of individual astrocytes as well as the percent of GFAP+ astrocyte coverage/image were assessed using the particle analysis plugin.

### 2.11. Assessment of Myelin Integrity

Analysis of intact myelin fibers and myelin debris was performed as previously described [22], using the Analyze particle plugin in FIJI/ImageJ [32] with size and circularity parameters for object differentiation. The parameters used to determine myelin debris were circularity = 0.3–1.0 and particle size = 0.5–10 µm^2^. To assess myelin fibers the analysis settings were as follows, circularity = 0.0–0.1 and particle size = 25-infinity µm^2^. The average total area covered by intact myelin fibers or myelin debris was quantified for each image and averaged for each animal.

### 2.12. Quantification of Protein Expression

Tissue processing and molecular assessments were done as reported previously [22,25]. Briefly, tissue from the right lateral neocortex and thalamus was homogenized in solubilization buffer (150 mM NaCl, 50 mM Tris pH 8.0, 1% Triton-X) and protease inhibitor cocktail (AEBSF 10.4 mM, Aprotinin 8 μM, Bestatin 400 μM, E-64 140 μM, Leupeptin 8 μM, Pepstatin A 150 μM, Cat#: P8340, Sigma, Saint Louis, MO, USA). Protein concentration was determined using a bicinchoninic acid assay in accordance with manufacturer’s instructions (Cat#23225; ThermoFisher) and quantified on a PHERAstar Spectrophotometer (BMG Labtech, Cary, NC, USA).

For assessment of cytokine expression, cortical and thalamic protein homogenates were sent to Quansys Bioscience (Logan, UT, USA) for cytokine analysis of rat IL-1a, IL-1b, IL-2, IL-4, IL-6, IL-10, IL-12, IFNy, and TNFa. Three experimental replicates were run for each sample and the means of the replicates were used for each animal. Cytokine concentration (in pg) were normalized to total protein concentrations (in mg) for each sample. Samples in which there was no detectable amount of cytokine were set to 0 pg/mg for analysis after verifying acceptable (>0.05 mg/mL) total protein concentrations.

To analyze myelin basic protein (MBP) expression, Western blotting was performed. Protein (5 μg) was boiled for 10 min in 2× Laemmli loading buffer and run at 200 volts for 30 min on Mini-PROTEAN TGX Stain-free 4–20% precast polyacrylamide gels (Cat #4568096; BioRad, Hercules, CA, USA). Protein was transferred onto 0.2 μm PVDF membranes using a Transblot Turbo transfer system (Bio-Rad) under the low molecular weight manufacturer settings (2.5 Amps, 25 Volts for 5 min). Western blotting was done on an iBind flex apparatus (Invitrogen) using primary antibodies rat anti-myelin basic protein (1:1000, Cat #MAB386; Millipore Sigma, Burlington, MA, USA) and mouse anti-actin (1:4000, Cat #66009-1-Ig; Proteintech; Rosemont, IL, USA) followed by anti-rat-HRP secondary antibody (1:5000; Cat#112-035-003; Jackson Laboratories, West Grove, PA, USA) and anti-mouse-HRP secondary antibody (1:5000, Cat #115-035-003; Jackson Laboratories; West Grove, PA, USA). Chemiluminescent images were taken on a ChemiDoc imaging system (BioRad, Hercules, CA, USA). Densitometric analysis was done in FIJI ImageJ [32] for actin and for each MBP isoform. Total MBP expression was measured by taking the densities of each individual MBP isoform band and adding them together to calculate the total sum. This method reduced biasing of the analysis by the variable degrees of white space between the MBP isoform bands in each run. MBP was then normalized to actin (the loading control), a consistent naive sample that was run on every blot, and to the average normalized density of sham controls. All Western blots were run in triplicates on three separate gels to reduce run-to-run variability potentially biasing the results.

### 2.13. Statistical Analysis

A Shapiro–Wilk test for normality of the data was done prior to statistical analysis. The number of animals to be assessed for each group was determined by power analysis using previous data, an alpha = 0.05 and a power of 80%. For data with a normal distribution One-way, Two-way, Three-way, or repeated measures ANOVA were done. For ANOVAs between and within group degrees of freedom, F and *p* values are reported. Data without a normal distribution were analyzed with a Kruskal–Wallis non-parametric test. Tukey post hoc corrections for multiple comparisons were done and statistical significance was set to a *p* value < 0.05. Data are presented as mean ± standard error of the mean (SEM) unless otherwise indicated.

## 3. Results

### 3.1. Physiology Was Not Impacted by Bup-SR-Lab Treatment

Body temperature was maintained at 37 °C with a feedback loop thermoregulatory system connected to a rectal thermometer to avoid potential confounds of hypothermia throughout all groups. As drug treatment was both randomized and blinded, there was little possibility to fully match injury metrics between treatment groups, however, there was no difference in pre-injury weight (One-way ANOVA F_3,20_ = 0.166, *p* = 0.918), injury intensity (One-way ANOVA F_3,20_ = 1.659, *p* = 0.227), or injury duration (One-way ANOVA F_3,20_ = 2.344, *p* = 0.157) between groups (Table 1).

To evaluate the potential effects of Bup on physiology at 4w post-injury blood oxygen saturation and heart rate were assessed prior to injury and at 1 h and 4w following cFPI. There was no discernable difference in blood oxygenation (Repeated Measures ANOVA F_3,19_ = 0.137, *p* = 0.936) or heart rate (Repeated Measures ANOVA F_3,19_ = 0.328, *p* = 0.805) between animal groups at any time point assessed (Figure 1A,B). Intracranial pressure at 4w post-cFPI was also comparable between treatment groups (Figure 1C) where the ICP readings were slightly lower in cFPI groups compared to sham injured groups regardless of drug treatment, but this was not significant (Two-Way ANOVA Drug: F_1,19_ = 0.044, *p* = 0.836; Injury: F_1,19_ = 0.638, *p* = 0.434).

Animal weight was also assessed prior to injury as well as at 1d, 2d, 3d, 1w, 2w, 3w, and 4w following cFPI to explore potential effects of Bup on post-injury weight retention. There were no significant changes in weight loss across all groups at any post-injury time point (Repeated measures ANOVA F_3,19_ = 0.537, *p* = 0.662). A dip in weight was observed in each group within the first week following sham or cFPI (Figure 1D) but none of the post-injury weights were significantly different from the animal’s pre-injury weight. All animal groups were able to recover from the weight loss and most animals were able to gain more weight than their starting pre-injury weights.

### 3.2. Neuronal Damage

As has been well characterized previously, cFPI did not generate gross tissue pathology. Neither saline nor Bup treated animals demonstrated indications of contusion, hematoma formation, or overt cortical compression. As the cFPI model induces diffuse neuronal pathology in the lateral neocortical and thalamic domains paired with our previous findings of Bup-related acute changes in the neocortex and thalamus, these regions of interest were the focus of all subsequent pathological assessments [22,34,36,37,38]. Overt neuronal damage/death and/or loss was assessed using H&E staining paired with size exclusion to eliminate the glial cell populations. Any cell over 30 μm^2^ was considered to be a neuron, and any neuron that demonstrated a heterochromatic nucleus and eosinophilic cytoplasm was considered to be damaged and/or undergoing early stages of cell death [22,34,39]. The total number of neurons was consistent across groups, with significantly higher cell numbers in the thalamus as compared to the cortex in all groups (Figure 2A; Three-Way ANOVA Region: F_1,40_ = 101.677, *p* = 1.52 × 10^−12^). There were very few damaged neurons in either the cortex or the thalamus following injury (Figure 2C; Kruskal–Wallis X^2^(3) = 0.941, *p* = 0.815), however, there was a significantly higher percentage of damaged neurons in the cortex compared to the thalamus regardless of injury or treatment group (Figure 2B; Kruskal–Wallis X^2^(1) = 20.39, *p* = 6.0 × 10^−6^). Neuronal cell size was also significantly larger in the cortex compared to the thalamus in all groups (Figure 2C; Three-Way ANOVA Region: F_1,40_ = 52.696, *p* = 8.19 × 10^−9^). There was a significant impact of injury on neuronal size in both regions (Three-Way ANOVA Injury: F_1,40_ = 101.12.495, *p* = 0.001; Cortex Two-Way ANOVA Injury: F_1,20_ = 7.19, *p* = 0.014; Thalamus Two-Way ANOVA Injury: F_1,20_ = 5.37, *p* = 0.031), with injured groups demonstrating significantly smaller neurons indicative of potential atrophy. There were no interactions between region, injury, and treatment for any metric.

Additionally, as somatic neuronal damage, in the form of neuronal membrane disruption, is present up to 4w following diffuse TBI [27], we investigated the potential effects of Bup-SR-Lab on neuronal membrane disruption using a cerebroventricular infusion of fluorescently tagged 10 kDa dextran, which is normally excluded from intact membranes. While there was a significant effect of injury in neuronal membrane disruption (Two-way ANOVA injury F_1,17_ = 11.76, *p* = 0.003), there was no difference between saline and Bup-SR-Lab treated groups regardless of injury, indicating that Bup did not alter this pathology at 4w post-injury (Figure 3; Two-way ANOVA treatment F_1,17_ = 0.71, *p* = 0.411).

### 3.3. Cytokine Changes Chronically following Bup-SR-Lab Administration

To assess potential chronic inflammatory responses in the brain following cFPI and/or Bup-SR-Lab administration, cytokine protein levels were assessed for each region (cortex and thalamus) in saline and Bup-SR-Lab treated animals at 4w post-sham or cFPI (Table 2). Levels of IL-2 were below detectable limits for nearly all samples tested regardless of treatment or injury group. However, concentrations of IL-1a, IL-1b, IL-4, IL-6, IL-10, IL-12, IFNy, and TNFa were measurable for most samples tested. There was no discernable treatment, region, or injury effect for IL-1b, IL-12, IFNy, or TNFa expression. There did appear to be a significant regional difference for IL-1a (Three-way ANOVA: Region: F_1,40_ = 4.586, *p* = 0.038) with higher cytokine expression in the thalamus compared to the cortex and a significant injury effect for both IL-6 (Three-way ANOVA Injury: F_1,40_ = 8.201, *p* = 0.007) and IL-10 (Three-way ANOVA Injury F_1,40_ = 4.435, *p* = 0.042) with lower cytokine expression following cFPI compared to sham-injured groups. There was a significant treatment difference for IL-4 (Three-way ANOVA Treatment: F_1,40_ = 5.34, *p* = 0.026) with higher cytokine expression following Bup-SR-Lab treatment compared to saline controls.

### 3.4. Microglial Morphology Is Altered Chronically following Bup-SR-Lab Administration

In our previous investigation we found that Bup-SR-Lab treatment resulted in region-specific changes in microglial and astrocyte morphologies [22]. To investigate these more subtle changes in neuroinflammation that might occur chronically following cFPI and Bup-SR-Lab treatment, microglia and astrocyte morphologies were analyzed. Activated microglia typically display smaller cell area, fewer total processes, reduced process endpoints and lower complexity per cell [22,35,38,40]. Alterations of microglia morphology were assessed at 4w following sham or cFPI and saline or Bup-SR-Lab treatment within the cortex and thalamus.

There were significantly more Iba-1+ microglia following cFPI (Three-Way ANOVA injury: F_1,48_ = 380.051, *p* < 0.001; Two-Way ANOVA Cortex: F_1,24_ = 6.874, *p* = 0.016; Thalamus: F_1,24_ = 6.533, *p* = 0.019) with a significant effect of region (Three-Way ANOVA region: F_1,48_ = 18.76, *p* < 0.001) in which there were higher numbers of microglia in the thalamus compared to the cortex, but no interactions between region and injury (Figure 4A; Three-Way ANOVA region x injury: F_1,48_ = 0.043, *p* = 0.84). The average size of microglia was also significantly lower following cFPI (Three-Way ANOVA injury: F_1,48_ = 11.061, *p* = 0.002; Two-Way ANOVA Cortex: F_1,24_ = 0.4, *p* = 0.534; Thalamus: F_1,24_ = 18.711, *p* < 0.001) with the thalamus showing significant injury effects. There was a significant interaction between the region and injury in regard to microglial cell size (Three-Way ANOVA region X injury: F_1,48_ = 5.64, *p* = 0.02). Neither number of microglia or cell size was effected by Bup-SR-Lab treatment (Figure 4A,B).

Microglia within the cortex of saline control animals following sham injury demonstrated over 150 processes/cell (153.29 ± 2.18) and almost 70 process endpoints/cell (68.67 ± 0.95), with a maximum process segment length of around 10 µm (9.92 ± 0.05 µm) and an average cell complexity index of >2 (2.22 ± 0.004 arbitrary units). Region had a significant effect on all morphological measurements except for the complexity index wherein microglia had fewer processes, less process endpoints and shorter maximum process lengths compared to microglia found within the cortex (Figure 4C–F; Three-Way ANOVA Region, Process number/microglia: F_1,31169_ = 682.518, *p* < 0.001; Endpoint number/microglia: F_1,31161_ = 739.123, *p* < 0.001; Maximum process length/microglia: F_1,31161_ = 131.589, *p* < 0.001; Complexity Index: F_1,31161_ = 2.438, *p* = 0.118). As has been shown previously [22], injury had a significant effect on microglial morphology in which microglia within the cortex or thalamus of post-cFPI animals had reduced process numbers, fewer process endpoints, and lower complexity indices (Three-Way ANOVA Injury, Process number/microglia: F_1,31161_ = 825.403, *p* < 0.001; Endpoint number/microglia: F_1,31161_ = 1025.477, *p* < 0.001; Maximum process length/microglia: F_1,31161_ = 46.347, *p* < 0.001; Complexity Index: F_1,31161_ = 251.795, *p* < 0.001). These changes with injury were consistent in both the cortex and the thalamus (Figure 4). There was a significant interaction between region and injury for the number of processes/microglia, the number of endpoints/microglia, and the complexity index (Three-Way ANOVA Region X Injury, Process number/microglia: F_1,31161_ = 203.572, *p* < 0.001; Endpoint number/microglia: F_1,31161_ = 233.443, *p* < 0.001; Complexity Index: F_1,31161_ = 46.876, *p* < 0.001).

Treatment with a single dose of Bup-SR-Lab at 15 min post-surgery also had a significant impact on the morphology of microglia 4w following injury (Three-Way ANOVA Treatment, Process number/microglia: F_1,31169_ = 143.789, *p* < 0.001; Endpoint number/microglia: F_1,31161_ = 139.427, *p* < 0.001; Complexity Index: F_1,31161_ = 30.936, *p* < 0.001). However, the maximum process length was not impacted by Bup-SR-Lab treatment (Three-Way ANOVA Treatment, Maximum process length/microglia: F_1,31161_ = 1.896, *p* = *0*.168). The complexity index was reduced in both the cortex and thalamus following cFPI with Bup treatment compared to their saline treated cFPI counterparts (Two-Way ANOVA Treatment Cortex: F_1,14374_ = 14.683, *p* < 0.001; Thalamus: F_1,16787_ = 17.08, *p* < 0.001). Microglia within the cortex, however, appear more sensitive to Bup-associated morphological alterations, having fewer processes and endpoints/cell compared to the saline-treated controls following either sham or cFPI (Figure 4; Two-Way ANOVA Treatment Cortex Process Number: F_1,14374_ = 185.933, *p* < 0.001; Endpoint Number: F_1,14374_ = 185.709, *p* < 0.001). These Bup-associated morphological changes in process number and endpoint number were not seen in the thalamus. There was a significant interaction between the treatment and injury for process number, endpoint number, and complexity index (Three-Way ANOVA Treatment X Injury, Process number/microglia: F_1,31161_ = 50.186, *p* < 0.001; Endpoint number/microglia: F_1,31161_ = 67.834, *p* < 0.001; Complexity Index: F_1,31161_ = 62.13, *p* < 0.001). This interaction between treatment and injury was found for all morphological metrics within the cortex (Two-Way-ANOVA cortex, Process number: F_1,14374_ = 55.21, *p* < 0.001; Endpoint number: F_1,14374_ = 62.38, *p* < 0.001; Max process length: F_1,14374_ = 19.29, *p* < 0.001; Complexity index: F_1,14374_ = 11.77, *p* < 0.001), and for the number of endpoints (F_1,16787_ = 8.54, *p* = 0.003), maximum process length (F_1,16787_ = 18.12, *p* < 0.001), and complexity index (F_1,16787_ = 59.02, *p* < 0.001) within the thalamus. There were also significant interactions among treatment, injury, and region for all morphological assessments (Three-Way ANOVA Treatment X Injury X Region, Process number/microglia: F_1,31161_ = 25.788, *p* < 0.001; Endpoint number/microglia: F_1,31161_ = 22.206, *p* < 0.001; Maximum process length: F_1,31161_ = 37.406, *p* < 0.001; Complexity Index: F_1,31161_ = 11.003, *p* < 0.001).

### 3.5. Astrocyte Changes Chronically following Bup-SR-Lab Administration

Similar to what we found in the microglial population, cFPI had a significant effect on astrocytes (Figure 5). There were greater numbers of astrocytes (Three-way-ANOVA Injury: F_1,40_ = 30.871, *p* < 0.001) with larger cell sizes (Three-way-ANOVA Injury: F_1,40_ = 10.763, *p* = 0.002), covering a greater percentage of the image (Three-way-ANOVA Injury: F_1,40_ = 55.36, *p* < 0.001) following cFPI. Injury also had an effect on basic astrocyte morphology, with injured astrocytes being more circular (Three-way-ANOVA Injury: F_1,40_ = 39.909, *p* < 0.001). The injury-induced astrocyte changes were more apparent in the thalamus compared to the cortex, with regional effects on astrocyte cell number (Three-way-ANOVA Region: F_1,40_ = 4.405, *p* = 0.042) and the percent of image covered (Three-way-ANOVA Region: F_1,40_ = 9.301, *p* = 0.004). There was a significant interaction between injury and region for astrocyte number (Three-way-ANOVA Injury X Region: F_1,40_ = 7.618, *p* = 0.009), percent of astrocyte coverage/image (Three-way-ANOVA Injury X Region: F_1,40_ = 9.244, *p* = 0.004), and cell circularity (Three-way-ANOVA Injury X Region: F_1,40_ = 4.474, *p* = 0.041), however, there was no interaction for astrocyte cell size (Three-way-ANOVA Injury X Region: F_1,40_ = 0.841, *p* = 0.365). When assessing the cortex, only injury had an impact on astrocytes (Two-way-ANOVA Cortex Injury, Percent area: F_1,20_ = 10.885, *p* = 0.004; Circularity: F_1,20_ = 7.03, *p* = 0.015). Astrocytes within the thalamus, however, appeared more sensitive to injury (Two-way-ANOVA Thalamus Injury, Cell number: F_1,20_ = 39.728, *p* < 0.001; Average astrocyte size: F_1,20_ = 7.428, *p* = 0.013; Percent area: F_1,20_ = 49.449, *p* < 0.001; Circularity: F_1,20_ = 47.789, *p* < 0.001). Thalamic astrocytes also were affected by Bup-SR-Lab treatment while their cortical counterparts were not (Figure 5) in which the astrocytes covered less area following cFPI in the Bup-SR-Lab treated group compared to the cFPI saline treated control (Two-way-ANOVA Thalamus Treatment, Percent area: F_1,20_ = 4.711, *p* = 0.042). There was a significant interaction between injury and treatment for the percentage covered by astrocytes (Two-way-ANOVA Thalamus Injury X Treatment: F_1,20_ = 4.599, *p* = 0.044) and astrocyte circularity (Injury X Treatment Three-Way ANOVA: F_1,40_ = 6.453, *p* = 0.015; Two-way-ANOVA Thalamus: F_1,20_ = 6.616, *p* = 0.018) in the thalamus.

### 3.6. Treatment with Bup-SR-Lab Altered Myelin Fibers in an Injury and Region-Specific Manner at 4 Weeks Post-Injury

The effects of Bup on myelin pathology were assessed at 4w following cFPI or sham injury (Figure 6) using immunohistochemical labeling for Myelin Basic Protein (MBP), as has been done previously [22]. There was no significant difference in the total number of myelin debris (Figure 6) between region (Three-way-ANOVA F_1,38_ = 0.497, *p* = 0.485), treatment (Three-way-ANOVA F_1,38_ = 0.596, *p* = 0.445), or injury group (Three-way-ANOVA F_1,38_ = 0.029, *p* = 0.866). There was, however, a significant regional difference with the cortex having more myelin fibers than thalamus (Three-way-ANOVA Region: F_1,38_ = 32.976, *p* < 0.001). Injury also impacted the total number of intact myelin fibers in which cFPI groups demonstrated reduced fiber numbers compared to sham injury groups with the highest number of myelin fibers being found in the sham-injured cortex of both saline and Bup-SR-Lab treated animals (Three-way-ANOVA Injury F_1,38_ = 8.724, *p* = 0.005; Two-Way ANOVA Cortex Injury: F_1,18_ = 5.155, *p* = 0.036; Two-Way ANOVA Thalamus Injury: F_1,20_ = 3.40, *p* = 0.08). The number of intact myelin fibers in the cortex of the sham injured saline treated group was significantly higher compared to the thalamus in all groups regardless of injury or treatment (One-way-ANOVA, F_7,38_ = 6.508; *p* = 4.5 × 10^−4^; vs. thalamus sham saline: *p* = 0.038; vs. thalamus sham Bup: *p* = 0.0031; vs. thalamus cFPI saline: *p* = 0.001; vs. thalamus cFPI Bup: *p* = 0.001). The number of intact myelin fibers was also significantly higher in the cortex of sham injured animals treated with Bup-SR-Lab compared to the thalamus of cFPI animals treated with either saline (*p* = 0.011) or Bup-SR-Lab (*p* = 0.012). There was no significant influence of Bup-SR-Lab treatment on myelin fiber number (Three-way-ANOVA Treatment: F_1,38_ = 0.927, *p* = 0.342).

As the different MBP isoforms are linked to different developmental stages of myelination, expression of the individual MBP isoforms (21.5 kDa, 18.5 kDa, 17.2 kDa, and 14.0 kDa) were also analyzed. There were significant regional differences in the overall expression of MBP and across each individual MBP protein isoform at 4w post-injury in which MBP expression was higher in the thalamus compared to the cortex (Figure 7; Appendix A; Three-Way ANOVA Overall: F_1,40_ = 17.83, *p* < 0.001; 21.5 kDa: F_1,40_ = 18.29, *p* < 0.001; 18.5 kDa: F_1,40_ = 18.525, *p* < 0.001; 17.2 kDa: F_1,40_ = 14.64, *p* < 0.001; 14.0 kDa: F_1,40_ = 12.01, *p* = 0.001). When comparing all injury and treatment groups in both regions there were significant differences in the overall expression of MBP (One-way-ANOVA F_7,40_ = 3.759, *p* = 0.003), expression of the 21.5 kDa isoform (One-way-ANOVA F_7,40_ = 3.771, *p* = 0.003), expression of the 18.5 kDa isoform (One-way-ANOVA F_7,40_ = 3.872, *p* = 0.003), expression of the 17.2 kDa isoform (One-way-ANOVA F_7,40_ = 3.356, *p* = 0.003), and expression of the 14.2 kDa isoform (One-way-ANOVA F_7,40_ = 2.703, *p* = 0.022). The overall expression of MBP was significantly higher in the thalamus of cFPI animals treated with Bup-SR-Lab compared to the cortex of cFPI animals treated with either saline (*p* = 0.043) or Bup-SR-Lab (*p* = 0.036). Expression of the 21.5 kDa isoform was significantly higher in the thalamus of sham animals treated with Bup-SR-Lab compared to the cortex of Bup-SR-Lab treated sham (*p* = 0.023) or cFPI (*p* = 0.019) animals as well as the cortex of cFPI animals treated with saline (*p* = 0.020). There was significantly higher expression of the 18.5 kDa isoform in the thalamus of Bup-SR-Lab treated animals regardless of injury compared to the cortex of cFPI animals treated with either saline (thalamus-sham-Bup *p* = 0.05; thalamus-cFPI-Bup *p* = 0.041) or Bup-SR-Lab (thalamus-sham-Bup *p* = 0.027, thalamus-cFPI-Bup *p* = 0.020). Expression of the 17.2 kDa isoform was significantly higher in the thalamus of cFPI animals treated with Bup-SR-Lab compared to the cortex of cFPI animals treated with either saline (*p* = 0.031) or Bup-SR-Lab (*p* = 0.029). There were no significant differences between any group for the 14.2 kDa isoform when Tukey post hoc corrections were done. There were also significant interactions between region and treatment in the overall expression of MBP primarily instigated by expression changes of the 18.5 kDa and 17.2 kDa isoforms (Three-way-ANOVA Region X Treatment; Overall: F_1,40_ = 4.441, *p* = 0.041; 18.5 kDa: F_1,40_ = 4.749, *p* = 0.035; 17.2 kDa: F_1,40_ = 4.363, *p* = 0.043) with higher MBP expression in the thalamus of Bup-SR-Lab treated groups regardless of injury.

### 3.7. Post-Injury Somatosensory Hypersensitivity Is Not Altered by Bup-SR-Lab Treatment

Since damage to the somatosensory cortex is correlated to changes in response to whisker stimulation, the whisker nuisance task (WNT) was performed pre-injury and at 4 weeks after sham or cFPI to investigate any potential alterations following Bup-SR-Lab administration (Figure 8). Animals sustaining cFPI displayed increased agitation in response to whisker stimulation, regardless of drug treatment at 4w post-injury compared to pre-injury (Figure 8A,B; Paired *t*-test cFPI: t(9) = −3.453 *p* = 0.007; Sham = t(8) = −0.759 *p* = 0.470), which is indicative of somatosensory sensitivity. While animals sustaining cFPI displayed hypersensitivity to somatosensory whisker stimulation as compared to animals sustaining sham injury (Two-Way ANOVA Injury: F_1,20_ = 7.136 *p* = 0.015), there were not significant effects of Bup-SR-lab treatment on post-injury WNT score (Two-Way ANOVA Treatment: F_1,20_ = 0.115 *p* = 0.738).

## 4. Discussion

The semi-synthetic opioid, Bup, is used for pain management and opioid addiction, however, there are limited studies on the potential effects of Bup following CNS injuries. We previously found Bup effects on acute TBI-induced pathology [22], therefore, the current study sought to evaluate the more chronic effects of Bup-SR-Lab on physiological, pathological, and behavioral alterations following a diffuse cFPI in adult male rats. Similar to what we had seen in our previous acute study [22], there were no physiological changes at 4w following cFPI. We had previously observed significant benefit regarding weight retention at 1d following cFPI and Bup-SR-Lab treatment [22]. While there was a trend toward higher weight retention in the Bup-SR-Lab treated groups at 1d post sham or cFPI in the current study, this effect was not significant and was not maintained at subsequent time points (Figure 1). Other studies investigating the effects of Bup on rodent models have also found varying impacts on weight. One study found weight loss to be more severe following Bup-SR-Lab treatment compared to saline controls [41], while another found that weight loss was slightly reduced following Bup treatment [42]. There have also been studies that, like the current study, found no weight-related differences in rats treated with Bup-SR compared to controls [17,18], demonstrating the variation in weight-related results.

The current study substantiates our previous findings [22] that cFPI does not lead to cell loss following injury alone at 4w post-injury (Figure 2). While we did observe regional differences in the number of total neurons and in the percentage of neurons considered damaged, neither of these metrics were linked to injury or Bup-SR-Lab treatment. Consistent with our and others previous investigations, neuronal membrane disruption was affected by injury, in which a greater percentage of neurons were membrane disrupted following cFPI [27,33,43,44]. There was also an effect of injury on cell size, in which cells were smaller following cFPI regardless of Bup-SR-Lab treatment. This is consistent with other findings of neuronal atrophy following cFPI from both our lab and others [25,37,45]. Most of these studies had been focused on the lateral neocortex, but the current study shows that the thalamus also undergoes a degree of neuronal atrophy following cFPI that does not appear to result in cell loss by 4w post-injury.

We found significant changes in cytokine expression in the current study, that we did not see in our acute study [22]. While we found lower expression of IL-4 and IL-10 in thalamus compared to the cortex in our 1d study, there were no Bup-associated effects at 1d post-cFPI [22]. At 4w, however, both IL-6 and IL-10 were reduced in cFPI compared to sham (Table 2). Expression of IL-6, which can be either pro or anti-inflammatory depending on the signaling mechanism and pathways activated, is typically very low in the healthy brain [46]. Our current study results are contrary to previous studies findings of increased IL-6 and/or IL-10 following fluid percussion injury. Following moderate-severe fluid percussion Zhu et al. found higher levels of IL-6, IL-1b, and TNF-alpha in the cortex of injured rats compared to sham controls [47]. Work done by Rowe et al. also found significant increases in IL-6 h following a cFPI with a trend toward increased IL-10 as well [48]. Another study done in pigs following a fluid percussion injury also found increased IL-6 in the CSF of injured pigs of both sexes as compared to sham controls [49]. All these studies investigated cytokine levels within hours of the injury. The more chronic timepoint of 4w in the current study could potentially account for the drastic difference in findings between our work and previous work following fluid percussion injury.

There was also a Bup-SR-Lab associated increase in IL-4 expression observed in the current study. There were no effects of injury on IL-4 expression, which is in accordance with previous studies [48], and there were no interactions between treatment and any other factor, therefore, the increase in IL-4 was primarily Bup-SR-Lab mediated (Table 2). This increase in IL-4 appeared more robust in the cortex, particularly within the sham injured groups. The cytokine, IL-4, is well known to promote anti-inflammatory phenotypes in microglia and astrocytes [50,51]. Upon treatment with IL-4, macrophages [52] and microglia [53,54] polarize to a more anti-inflammatory phenotype associated with increased expression of anti-inflammatory cytokines such as Ym1, Fizzl, and IL-10. Astrocyte expression of neurotrophic factors, such as BDNF and NGF, and anti-inflammatory cytokine production are also associated with IL-4 induction [51]. Induction with IL-4 also increased the expression of delta opioid receptors (DOR) on cultured BV2 microglia [54]. Agonist to the DOR similarly resulted in induction of an anti-inflammatory microglia phenotype even when primed with pro-inflammatory lipopolysaccharide or hypoxia [54]. Treatment with Bup in cultured macrophages also enhanced anti-inflammatory cytokine expression [52]. As Bup acts as a DOR antagonist the Bup-associated anti-inflammatory effects seem counterintuitive, however, DOR antagonism has also been shown to induce anti-inflammation reducing TNF-alpha expression and reducing microglial coverage [55].

Anti-inflammatory induction has been shown to alter microglial morphology to a more activated phenotype [53]. Upon activation (either via pro or anti-inflammatory mechanisms) the morphology of microglia changes. Activated microglia typically increase in number, cover a larger percent of region of interest [38,56], and usually have larger cell bodies with shorter processes and less complex process networks [40]. Four weeks following cFPI we found a trend toward more Iba-1+ microglial in both the cortex and thalamus regardless of Bup treatment, however, this was not significant. There was, however, a significant reduction in microglial size, process number, and endpoint number following cFPI, indicating microglial activation (Figure 4). This agrees with previous work by our group and others demonstrating morphological activation of microglia following cFPI [22,38,40,57] in both the cortex and the thalamus. Similar to our previous acute study, we also found regional differences in microglial morphology, however, in the current study we incorporated sham controls and therefore were able to look at general microglial morphological differences between cortex and thalamus. Thalamic microglia within the sham saline control groups had fewer overall processes, fewer end points, reduced maximum process lengths and smaller complexity indices as compared to microglia within the sham saline cortex. Interestingly, these regional differences are not present in the Bup-SR-Lab treated shams, in which the cortical microglia have significantly fewer processes, fewer process end points, and reduced maximum process length compared to microglia within the sham saline treated cortex (Figure 4). This is in alignment with our 1d study that demonstrated cortical microglia were more susceptible to Bup-induced morphological changes [22]. Following cFPI there was also a significant Bup-SR-Lab treatment effect on the number of processes and number of endpoints in the cortex that was not recapitulated in the thalamus, further indicating a cortical susceptibility to Bup-associated microglial changes. Complexity index, however, was significantly affected by Bup in both the cortex and thalamus 4w following cFPI, in which complexity was reduced. Microglial complexity was also found significantly reduced in the cortex of cFPI animals with Bup-SR-Lab treatment at 1d post-injury, however, at that acute time point there was no significant change in the thalamus [22].

Whereas microglia within the cortex appear to be susceptible to Bup-induced morphological changes, cortical astrocytes did not appear to be influenced by either injury or Bup-SR-Lab treatment (Figure 5). Astrocytes within the thalamus, however, demonstrated significantly more numbers, covering a larger percentage of the image, with larger and more circular cell areas in saline-treated cFPI compared to sham saline controls. While astrocytes within the thalamus of Bup-SR-Lab treated cFPI animals maintained increased numbers, percent coverage, and circularity over their sham Bup-SR-Lab treated counterparts, Bup treatment did extinguish the increase in astrocyte cell size in the thalamus 4w following cFPI. Astrocytes also covered less area following cFPI in the Bup-SR-Lab treated thalamus compared to the cFPI saline treated control. These data support our previous acute finding that astrocytes within the thalamus appear to be more susceptible to Bup-associated changes.

Other studies have also demonstrated region-specific differences in the astrocyte population. Astrocyte morphological, physiological, functional, and damage responses are heterogeneous across astrocytes within and between brain regions [58]. Astrocytes within the cortex and thalamus have been shown to develop at different rates, with cortical astrocytes developing following synaptic input stabilization and thalamic astrocytes reaching maturity prior to synaptic stabilization [59,60]. A study using single-cell RNA sequencing also demonstrated astrocyte heterogeneity within the cortex with variation across cortical layers [61]. Immunoreactivity for GFAP was also seen to vary between different brain regions in the human population [62]. Torres-Platas et al. found region-specific astrocyte alterations, with significant decreases in GFAP mRNA and protein in the thalamus but not the cortex of postmortem tissue from individuals that died by suicide compared to controls [62]. Another group also saw changes in T2-weighted and FLAIR hyperintensity in the thalamus, but not in the cortex of individuals suffering from astrocytophathy [63]. Bup treatment has also been shown to affect brain regions differently by other groups. Prenatal exposure to Bup resulted in higher GFAP levels specifically within the hippocampus [64]. Another group found a slight reduction in the percentage of GFAP+ area covered in the corpus callosum of mice days following Bup administration [42]. These regional Bup-associated differences could be associated with differences in expression of opioid receptors in the cortex compared to the thalamus. One group found a significant reduction in mu opioid receptor expression in the cortex and a drastic increase in delta opioid receptor expression in the thalamus of mice following a chronic compression injury that appeared to be region specific [65]. This same injury was shown to induce decreases in delta opioid receptor expression in both microglia and astrocytes without changes in glial mu opioid receptor expression [55]. Further, inhibition of both mu and delta opioid receptors reduced mechanical sensitivity in compression injured rats, implicating these receptors in injury-induced sensitivity [62]. Astrocyte specific knockouts for delta opioid receptors also showed higher mechanical sensitivity threshold to Von Frey stimulation in both males and females, indicating that astrocyte opioid expression may play a key role in mechanical allodynia [66].

Similar to our 1d study [22], the number of myelin debris stayed consistent throughout all groups at 4w post-injury. Debris cleanup is initiated by activated microglia starting at 2–3 weeks after induction of demyelination with cuprizone and by 4–5 weeks there was nearly complete clearance of myelin debris [67]. Based on this, it is possible that in the current 4w study nearly all myelin debris would have been cleared. It is also possible that the puncta considered to be myelin debris in this study could be cross-sections of myelinated axons, which would explain the consistency between injury groups in the thalamus, as the axons in this region are variably arranged. However, this would not explain the consistency between the thalamus and cortex, which has a linearly arranged fiber orientation, but similar numbers of myelin debris puncta in our study. Further studies are needed to explore this possibility.

We did, however, see injury and regional differences when assessing the number of myelin fibers in the current study. There were significantly more myelin fibers in the cortex compared to the thalamus of sham controls regardless of Bup-SR-Lab treatment. Injury reduced the fiber numbers in the cortex and was trending toward reduction in the thalamus, but this trend was not significant. When comparing our current results to our previous investigation of myelin fibers at 1d post-injury we found that there were greater numbers of intact myelin fibers in the cortex of cFPI animals at 4w compared to what we saw in our previous 1d cFPI study [22] regardless of treatment group. It was previously shown that demyelinated axons can remain viable and start to undergo remyelination between 3d and 1w post-injury in the corpus callosum of mice [68]. In the current study, 4w likely was enough time for potential remyelination which could explain the increase seen in myelin fiber numbers in the cortex of cFPI animals treated with saline or Bup-SR-Lab at 4w post-injury compared to our previous 1d finding [22]. Kappa (κ) opioid receptor (KOR) agonists have previously been implicated in regulating myelination by promoting and accelerating oligodendrocyte differentiation [69]. However, the potential role of KOR in cortical remyelination would need to be further explored as Bup is a KOR antagonist which should have resulted in a decrease in myelin fibers that we did not observe in the current study. The thalamus of cFPI animals had fewer intact myelin fibers at 4w post-injury compared to our previous observation in the thalamus at 1d post-cFPI [22] regardless of treatment group, suggesting a regional distinction following cFPI. A previous study investigating the degradation of myelin basic protein (MBP), an important protein involved in the process of myelination, found significant MBP breakdown in the hippocampus following TBI [70]. It is likely that breakdown of MBP is happening in other subcortical regions, such as the thalamus and might explain the decreased number of myelin fibers in the thalamus at 4w following cFPI, although this has not specifically been investigated.

While the numbers of myelin fibers were lower in the thalamus compared to the cortex, the overall expression levels of MBP were higher in thalamus compared to cortex, with the highest levels of MBP being expressed in the thalamus of Bup-SR-Lab treated animals. Interestingly, overall MBP expression and expression of all isoforms was comparable between the cortex and thalamus in sham saline treated animals, indicating that there are not general regional differences in terms of MBP expression. Injury did not appear to alter the MBP expression in either region, however, Bup-SR-Lab treatment did appear to increase expression of MBP in the thalamus most notably in the 21.5 kDa, 18.5 kDa, and the 17.2 kDa isoforms. Expression of the 21.5 kDa isoform in the thalamus of Bup-SR-lab treated sham animals was significantly elevated compared to the cortex of sham animals treated with Bup-SR-Lab and both cFPI groups and the cortex regardless of treatment group. At 4w post-cFPI we observed higher expression of the 17.2 kDa and 18.5 kDa isoforms in the thalamus compared to the cortex of cFPI animals in a Bup-dependent manner. These isoforms of MBP are thought to have regulatory effects in myelination. The 17.2 kDa isoform has been shown to be expressed at high levels in developing oligodendrocytes and may have roles in the radial component of mature myelin and the 18.5 kDa isoform is known to interact with membranes to form compact myelin and facilitate assembly of the cytoskeleton to adhere to membranes of oligodendrocytes [71]. Oligodendrocyte death has been connected to TBI and is suggested to be a hallmark of diffuse axonal injury and a previous study observed oligodendrocyte death from 2d to up to 4w after TBI injury [72]. Another study also observed a significant loss of mature oligodendrocytes within the external capsule with oligodendrocyte numbers returning to normal within 2w following cortical injury, indicating a potential for oligodendrocyte differentiation and remyelination by 4w post-injury [73]. The alterations that we see in the 18.5 kDa and 17.2 kDa isoforms of MBP at 4w post-injury could be a consequence of remyelination in the thalamus of Bup-treated animals following TBI, however, future assessments of oligodendrocyte number and development following cFPI and Bup-SR-Lab treatment would need to be done to investigate this possibility.

As sustained release Bup works to reduce pain for days [19,74] and alterations in opioid signaling in astrocytes alters mechanical sensitivity [66], the WNT was performed to investigate potential Bup-SR-Lab-associated changes in somatosensory sensitivity 4w following cFPI or sham injury. In agreement with previous studies from both our and other groups [24,28,30,31], animals sustaining cFPI displayed hypersensitivity and agitation to the stimulation as compared with animals sustaining sham injury at 4w post-injury. Treatment with Bup-SR-Lab, however, did not seem to have a significant effect on either sham or cFPI injured groups (Figure 7). A previous pharmacological magnetic resonance imaging study displayed reduced activation in cortical and subcortical regions including the somatosensory cortex and thalamus following high and low doses of Bup in rats [75]. As these regions are implicated in pain response and contain high densities of opioid receptors [76], the decreased activation seen in these brain regions would indicate a potential reduction in somatosensory sensitivity, however, that was not observed in the current study. Another study concluded that Buprenorphine had less effect on sensory information processing in the primary somatosensory cortex and had no effect on sensory-motor filtering [77]. Interestingly, a study investigating pain responses in IL-4 knockout mice found that mechanical sensitivity was higher without IL-4 but was not exacerbated by a subsequent injury [65]. As IL-4 expression was upregulated in our Bup-SR-Lab treated animals this could be a potential mechanism for the slight exacerbation in somatosensory sensitivity in the sham Bup-SR-Lab treated animals compared to saline-treated control sham animals (Figure 7).

## 5. Conclusions

The interplay between the use of analgesics, specifically that of Bup, and diffuse brain injuries in experimental TBI studies is widely unknown due to limited research. This study sought to determine the potential chronic physiological, pathological, and behavioral alterations 4w following TBI and treatment with Bup-SR-Lab. The findings of this study show that preclinical use of Bup-SR-Lab following diffuse cFPI influence glial alterations following injury in a region-specific manor. Further studies into the specific molecular mechanisms involved in the alterations in glial morphology, neuroinflammation, and cell damage are required in order to thoroughly understand the effects of Bup following diffuse TBI.

## Figures and Tables

**Figure 1 pharmaceutics-14-02068-f001:**
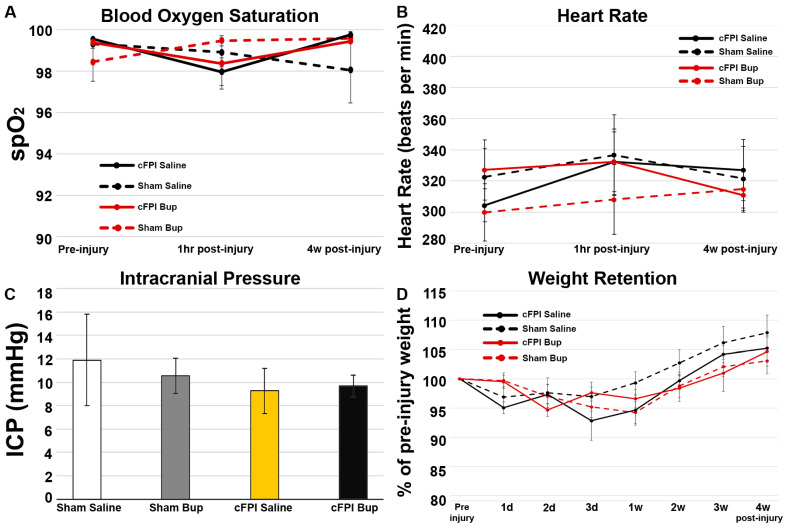
No significant changes in physiology or weight retention with Bup-SR-Lab (Bup) treatment following central fluid percussion injury (cFPI). Physiological readouts of (**A**) blood oxygen saturation and (**B**) heart rate in saline and Bup treated adult male rats pre-injury, 1 h, and 4w post-cFPI or sham injury. (**C**) Bar graph representing the average Intercranial pressure (ICP) of animals at 4w post sham or cFPI. (**D**) weight retention from pre-injury weight (100%) over the 4w time course. Animals in all groups lost weight following either sham or cFPI but were above their pre-injury weight by 4w post-injury and there were not differences between groups. n = 6 rats/group. Mean ± SEM.

**Figure 2 pharmaceutics-14-02068-f002:**
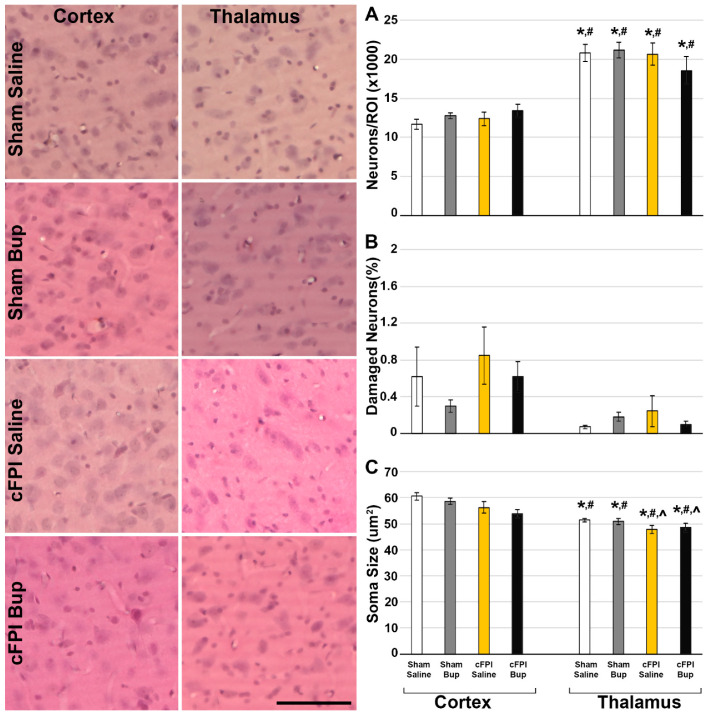
Cell number, damage, and size was not altered by Bup-SR-Lab (Bup) treatment at 4w post sham or central fluid percussion injury (cFPI). Representative photomicrographs of the cortex or thalamus from animals treated with saline or Bup following sham or cFPI and labeled with hematoxylin and eosin (H&E). (**A**–**C**) Corresponding bar graphs depicting (**A**) the total number of neurons/region of interest (ROI) in the thousands, (**B**) percent of total H&E stained neurons exhibiting pycnotic nuclei or eosinophilic cytoplasm indicative of damage for the cortex and thalamus, and (**C**) Soma size. n = 6 rats/group. Mean ± SEM. Scale = 100 μm. * *p* < 0.05 compared to sham saline cortex, # *p* < 0.05 compared to sham Bup cortex, ^ *p* < 0.05 compared to cFPI saline cortex.

**Figure 3 pharmaceutics-14-02068-f003:**
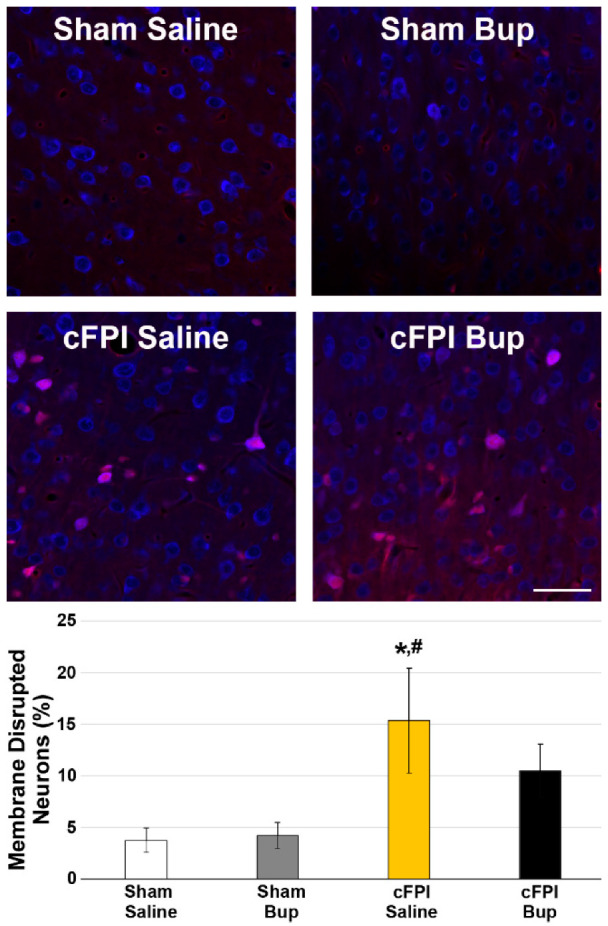
Treatment with Bup-SR-Lab (Bup) did not impact cortical neuronal membrane disruption. Representative photomicrographs of neuronal membrane disruption in the lateral neocortex of rats treated with saline or Bup. Cell uptake of the cell-impermeable 10 kDa fluorescently tagged dextran (red) indicated neurons sustaining membrane disruption. Neurons were visualized with NeuroTrace fluorescent Nissl stain (blue). Corresponding bar graph depicting the percentage of total neurons that demonstrated membrane disruption in saline and Bup treated animals following Sham or central fluid percussion injury (cFPI). The percentage of dextran + neurons was consistent between injury groups, indicating that buprenorphine treatment did not affect the percentage of membrane disrupted neurons in the cortex. n = 4–6 rats/group. Mean ± SEM. Scale = 50 μm. * *p* < 0.05 compared to sham saline cortex, # *p* < 0.05 compared to sham Bup cortex.

**Figure 4 pharmaceutics-14-02068-f004:**
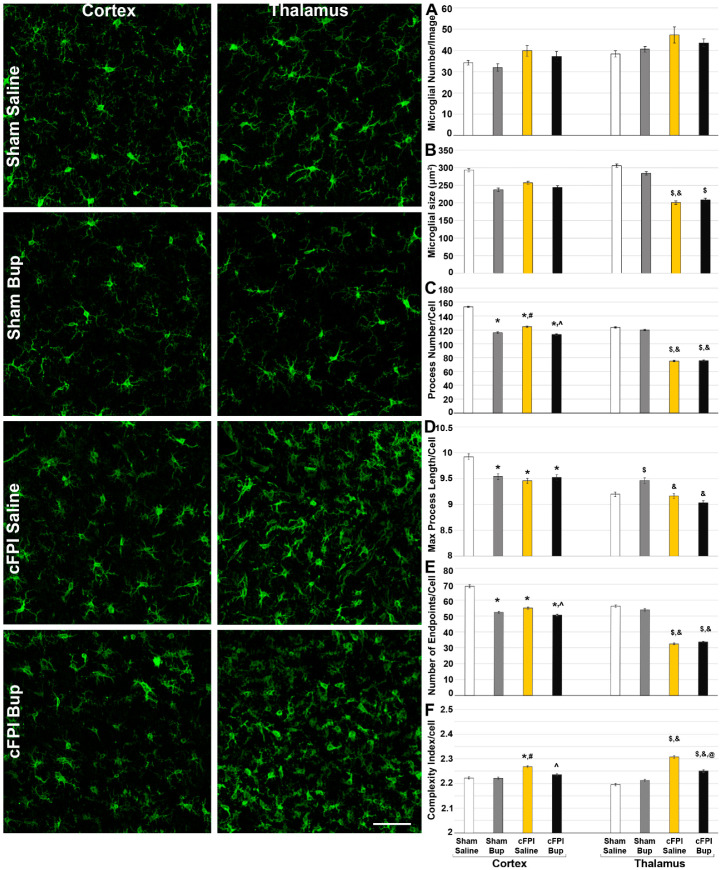
Bup-SR-Lab (Bup) treatment is associated with microglial morphologic alterations primarily within the cortex 4w following Bup-SR-Lab treatment. Representative photomicrographs of Iba-1 labeled microglia in the cortex and thalamus from rats 4w post-cFPI or sham and treated with saline or Bup. Bar graphs depicting microglial characteristics and morphology indicative of activation: (**A**) number of microglia/image analyzed, (**B**) average microglia cell size, (**C**) Number of processes/microglia, (**D**) average maximum process length/microglial cells, (**E**) number of process end points/cell, and (**F**) the average complexity index of the microglial process network. Microglial morphologies were significantly different in the cortex of sham control rats treated with Bup compared to cortical microglia in sham and saline treated control animals. Additionally, microglial process number, endpoint number, and process network complexity was altered in the cortex, but not the thalamus of Bup-treated cFPI rats compared to their saline-treated counterparts, suggesting regional specificity in the effects of Bup. n = 6 rats/group. Mean ± SEM. Scale = 50 μm. * *p* < 0.05 compared to sham saline cortex, # *p* < 0.05 compared to sham Bup cortex, $ *p* < 0.05 compared to sham saline thalamus, & *p* < 0.05 compared to sham Bup thalamus, ^ *p* < 0.05 compared to cFPI saline cortex, @ *p* < 0.05 compared to cFPI saline thalamus.

**Figure 5 pharmaceutics-14-02068-f005:**
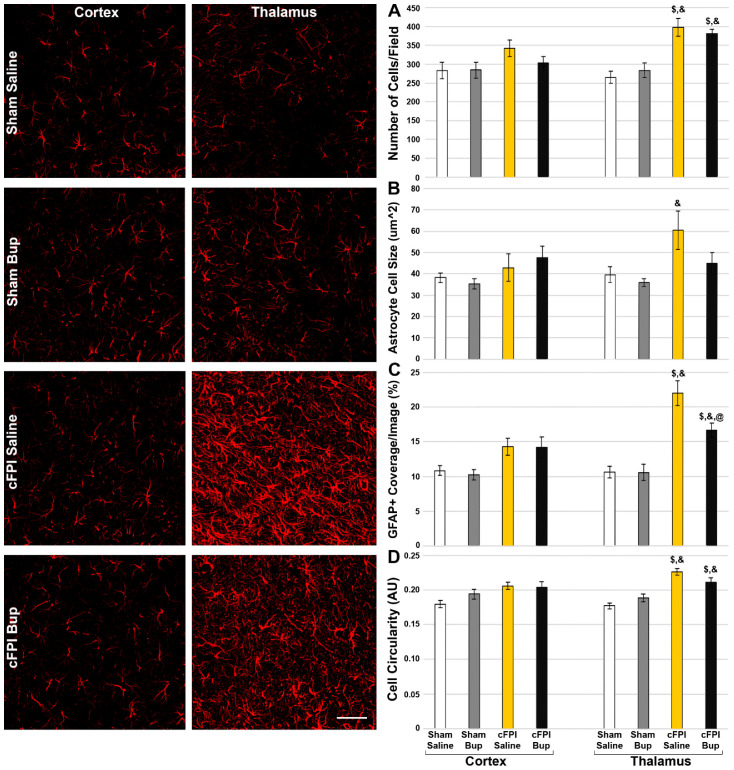
Treatment with Bup-SR-Lab (Bup) resulted in reduced astrocyte coverage in the thalamus. Representative photomicrographs of the astrocytic glial fibrillary acidic protein (GFAP) in the cortex and thalamus from rats 4w post-cFPI or sham and treated with saline or Bup. Bar graphs depicting astrocyte (**A**) cell number, (**B**) Average astrocyte cell size, (**C**) % of GFAP+ coverage/image, and (**D**) astrocyte cell circularity. While cortical astrocytes were not significantly changed following cFPI with or without Bup treatment, astrocytes within the thalamus demonstrated reduced circularity following cFPI and Bup. n = 6 rats/group. Mean ± SEM. Scale = 50 μm. $ *p* < 0.05 compared to sham saline thalamus, & *p* < 0.05 compared to sham Bup thalamus, @ *p* < 0.05 compared to cFPI saline thalamus.

**Figure 6 pharmaceutics-14-02068-f006:**
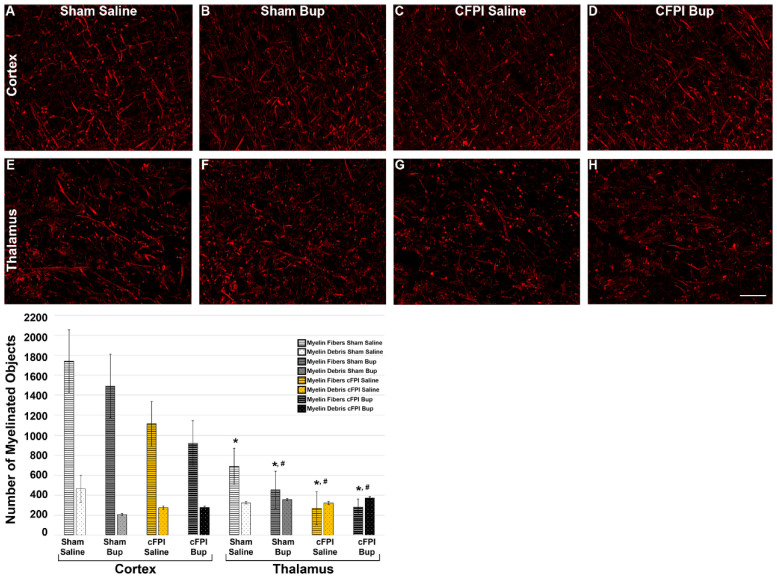
The number of myelin fibers is not significantly altered with cFPI or Bup-SR-Lab (Bup) treatment 4w post-injury. Representative photomicrographs of myelin basic protein (MBP) labeling in the (**A**–**D**) cortex and (**E**–**H**) thalamus of (**A**,**B**,**E**,**F**) sham or (**C**,**D**,**G**,**H**) cFPI animals treated with (**A**,**C**,**E**,**G**) saline or (**B**,**D**,**F**,**H**) Bup. Bar graph depicting the average number of MBP+ fibers (bars with horizontal lines) and MBP debris (bars with dots) in saline and Bup treated animals 4w after sham or cFPI. The number of myelinated debris was consistent among groups. There were fewer myelin fibers in the thalamus compared to the cortex, however, there was no Bup, or injury effect observed. n = 6 rats/group. Scale = 50 μm. * *p* < 0.05 compared to Cortex-Sham-Saline; injury differences: # *p* < 0.05 compared to Cortex-Sham-Bup. n = 6 rats/group. Mean ± SEM. * *p* < 0.05 compared to sham saline cortex, # *p* < 0.05 compared to sham Bup cortex.

**Figure 7 pharmaceutics-14-02068-f007:**
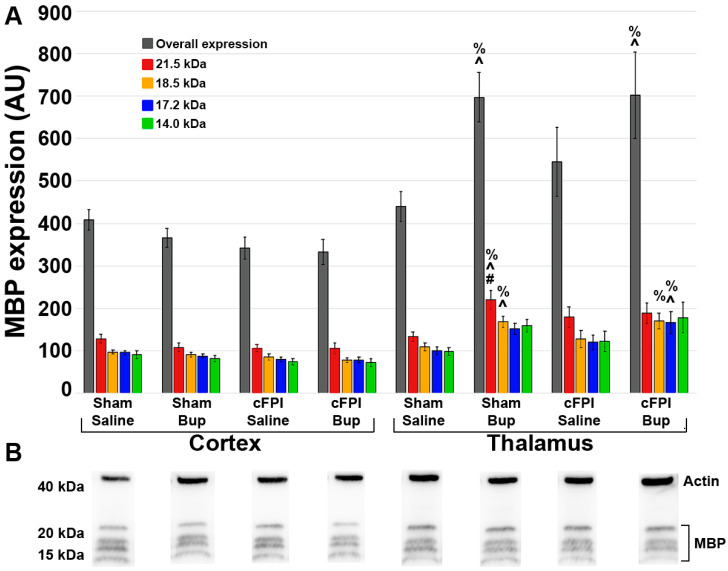
Expression of MBP was altered in a region-specific as well as a drug-and-region related manner at 4 weeks post-injury. (**A**) Bar graph depicting average overall MBP expression (gray bars), as well as expression of 21.5 kDa (red bars), 18.5 kDa (orange bars), 17.2 kDa (blue bars), and 14.0 kDa (green bars) MBP isoforms in the cortex and thalamus of saline and Bup treated animals following sham or cFPI injury. (**B**) Representative Western blots of actin (band at ~40 kDa) and MBP (bands at ~15–20 kDa). n = 6 rats/group. Mean ± SEM. # *p* < 0.05 compared to sham Bup cortex, ^ *p* < 0.05 compared to cFPI saline cortex, % *p* < 0.005 compared to cFPI Bup Cortex.

**Figure 8 pharmaceutics-14-02068-f008:**
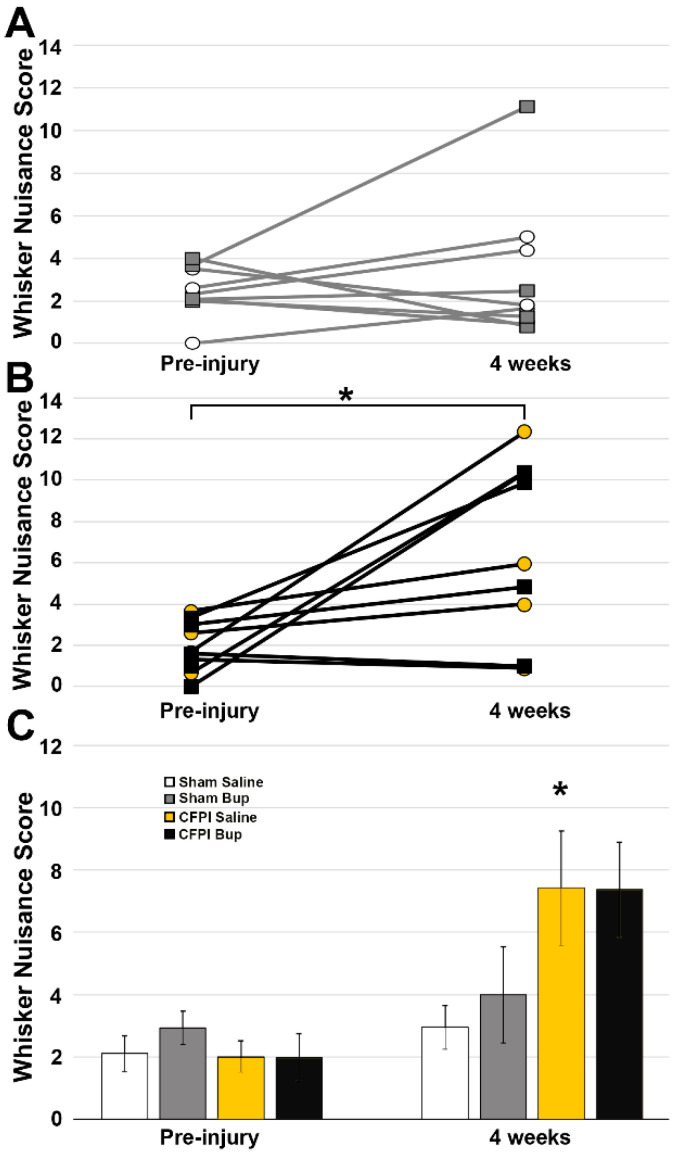
Somatosensory sensitivity was not impacted by Bup-SR-Lab (Bup) treatment. Line graphs of (**A**) sham or (**B**) cFPI animals whisker nuisance scores pre-injury and at 4w post-injury in groups treated with saline (circles; white circles in A are sham saline treated and yellow circles in B are cFPI saline treated animals) or Bup (squares; grey squares in A are sham Bup treated and black squares in B are cFPI Bup treated animals). Each line represents an individual animal. * *p* < 0.05 compared to pre-injury score. (**C**) Bar graph depicting the mean whisker nuisance task score prior to and 4w following sham or cFPI and saline or Bup treatment. n = 6 rats/group. Mean ± SEM. * *p* > 0.05 compared to sham saline at 4w.

**Table 1 pharmaceutics-14-02068-t001:** Physiological readouts in sham or injured and saline or buprenorphine (Bup) treated animals prior to and 4 weeks following central fluid percussion injury (cFPI).

	Sham Saline	Sham Bup	cFPI Saline	cFPI Bup
**Pre-injury Weight (g)**	482 (62)	489 (71)	498 (87)	513 (89)
**Injury Intensity (atm)**	Sham injury	Sham injury	2.08 (0.04)	2.05 (0.05)
**Injury duration (ms)**	Sham injury	Sham injury	20.97 (0.56)	21.43 (0.50)
**Recovery Time (min)**	62.67 (43.43)	48.33 (6.47)	60.83 (29.21)	72.50 (36.75)

g = grams, atm = atmospheric pressure, ms = milliseconds. n = 6 rats/group. Data presented as mean (standard deviation).

**Table 2 pharmaceutics-14-02068-t002:** Cytokine concentrations (pg/mL) in cortex and thalamus of saline and Bup-SR-Lab treated rats at 4w following Sham or cFPI.

	IL-1a	IL-1b	IL-4	IL-6	IL-10	IL-12	IFNy	TNFa
**Sham Saline cortex**	17.36 (6.01)	7.23 (1.56)	14.43 (2.37)	119.80 (29.20)	26.05 (1.62)	269.88 (21.10)	47.03 (8.09)	28.68 (6.95)
**Sham Bup Cortex**	22.73 (6.26)	9.51 (1.14)	28.02 (7.00)	158.28 (34.62) @	27.84 (1.87)	324.02 (32.19)	63.58 (12.54)	38.11 (4.35)
**TBI Saline Cortex**	17.08 (4.79)	9.22 (1.72)	18.50 (2.34)	111.41 (20.73)	22.81 (3.46)	291.84 (22.75)	56.51 (8.12)	42.77 (4.94) @
**TBI Bup Cortex**	14.13 (3.89)	7.95 (1.53)	19.44 (6.77)	83.14 (27.02)	21.16 (5.31)	276.89 (24.79)	36.04 (8.02) #	36.46 (5.12)
**Sham Saline Thalamus**	23.64 (4.52)	12.08 (3.92)	16.49 (7.06)	164.31 (44.03) @	26.57 (3.24)	330.17 (39.74)	54.74 (10.63)	37.51 (9.70)
**Sham Bup Thalamus**	27.91 (4.84)	11.27 (2.52)	22.54 (3.18)	165.38 (36.41) @	20.76 (2.77)	259.37 (25.48)	49.37 (4.95)	47.33 (5.86) *,@
**TBI Saline Thalamus**	25.14 (4.18)	9.85 (2.27)	15.45 (4.53)	28.48 (17.46)	19.44 (3.04)	314.33 (30.86)	45.86 (5.44)	23.97 (5.02)
**TBI Bup Thalamus**	26.28 (6.63)	14.88 (4.19) *	29.03 (5.68)	123.40 (39.34) @	17.64 (4.24) #	293.65 (22.03)	52.68 (12.08)	37.41 (7.78)

Data presented as mean (Standard Error of the Mean). * *p* < 0.05 compared to Sham Saline Cortex; # *p* < 0.05 compared to Sham Bup-SR-Lab Cortex; @ *p* < 0.05 compared to cFPI Saline Thalamus.

## Data Availability

The curated data presented in this study and the protocols used to generated this data are publicly available on the Open Data Commons for Traumatic Brain Injury.

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
