# Peer review of "Post-Injury Buprenorphine Administration Is Associated with Long-Term Region-Specific Glial Alterations in Rats"

_pharmaceutics, 2022, doi:10.3390/pharmaceutics14102068_

Round 1

Reviewer 1 Report

I find this paper very well written with clear presentation of data and sound experimental design. The authors present important data for the field, as many are required to use buprenorphine in experimental model management of pain etc. 

The paper also nicely shows brain-region dependent differences in rodents in sham and injury animals, confirming other reports, and also now with Bup treatment which is the main novel finding of the paper.

It is rare to have no further comments, but I recommend the paper published as is.

Author Response

Thank you so much for your review and your time.

Reviewer 2 Report

A very interesting manuscript on a rodent TBI model with focus on buprenorphine  approach on glial deficiencies.
Paper is of interest for the readers.
I think it will be a good addition to the journal in a very new and exciting area.
Still:
- ethic number should be provided , since this is quite aggressive to the model. not just some "regular local and international general rules were applied".  It will be better actually to provide the all ethical approval on the supplementary material.
Introduction is ok. Maybe some newer titles on TBI can be added: even from MDPI titles from the last months and less than a year. Those from 7 years ago are fine, but lets update the reference part. 
Other than that Discussion is very complex and nicely done.
Results are promising.
Conclusions are balanced.

Author Response

Thank you so much for your review and your time.

-Ethic number should be provided , since this is quite aggressive to the model. not just some "regular local and international general rules were applied".  It will be better actually to provide the all ethical approval on the supplementary material. 

We have now included the specific approval number for this study (line 92). This study was approved by the Institutional Animal Care and Use Committee (IACUC) at Virginia Commonwealth University, which is an AAALAC accredited institution.

Reviewer 3 Report

The research is very interesting, and the figures are convincing. Their method on morphological analysis based on Fiji is rational. The conclusion will help improve the treatment of TBI in the future. Here are my comments.

1 Line 220 Please state the method in detail with a citation of your previous study.

2 Please perform the skeleton analysis on astrocytes. I know there are overlaps between the processes, but this question can be approached through Fiji. Here is a paper for reference. Juvenile mild traumatic brain injury elicits distinct spatiotemporal astrocyte responses DOI: 10.1002/glia.23736

3 Assessment neurons of HE stain by Fiji is inaccurate. It would be nice to perform this analysis on fluorescence images of Neun IHC. Although there will be a decrease in size of degenerated neurons, other changes on the staining, processes, and shape should also be considered. Besides, neuron size will increase under cytotoxic edema. Many papers and books have described neuron pathology. I propose the study team evaluate neuron damage based on the biomarkers such as FJB, Tunel.

4 Figure 2. The background colour of the images is inconsistent. The overt pathological change is minor among the groups. It would be better to perform cell counting in lower magnification.

5 Figure 4. The cell complexity can be analyzed by FracLac, a convenient plugin of Fiji. It can measure the deformation of the cell outlines. Just a proposal.

6 Figure 7. Please provide the raw gel of wb as supplementary material.

Author Response

Thank you so much for your review and your time.

1 Line 220 Please state the method in detail with a citation of your previous study.

Thank you for catching this. We have now added in the methods used for the Dextran infusion with citations. Added in Lines 133-140 and highlighted purple.

2 Please perform the skeleton analysis on astrocytes. I know there are overlaps between the processes, but this question can be approached through Fiji. Here is a paper for reference. Juvenile mild traumatic brain injury elicits distinct spatiotemporal astrocyte responses DOI: 10.1002/glia.23736

 This is a very interesting paper and we had considered applying the same skeleton analysis we using for our microglia in our astrocyte assessments. This analysis, however, is still not well vetted for adult diffuse TBI studies. For the astrocyte studies, we tried to stay consistent with previous literature and with our previous study to make things easier to cross-assess once we make this data publicly available. We will definitely consider attempting to validate the skeleton analysis for GFAP+ astrocytes in the adult CFPI model in a future study.

3 Assessment neurons of HE stain by Fiji is inaccurate. It would be nice to perform this analysis on fluorescence images of Neun IHC. Although there will be a decrease in size of degenerated neurons, other changes on the staining, processes, and shape should also be considered. Besides, neuron size will increase under cytotoxic edema. Many papers and books have described neuron pathology. I propose the study team evaluate neuron damage based on the biomarkers such as FJB, Tunel.

We have done assessments using NeuN IHC paired with H&E as well as TUNEL on tissue from CFPI animals previously (doi.org/10.3389). We found that there was no TUNEL signal and, interestingly, that there was a consistent subpopulation of H&E-stained neurons (based on a size exclusion to eliminate glial cells) did not label with NeuN. For our subsequent studies, therefore, we have opted to use the H&E assessments to include this subpopulation.

4 Figure 2. The background colour of the images is inconsistent. The overt pathological change is minor among the groups. It would be better to perform cell counting in lower magnification.

We imaged the H&E at 20X magnification, which was too low of a magnification to see individual cells for the figure. The reviewer is correct that the stain is variable, therefore, we did background subtraction with a 50pxl rolling ball radius to reduce the stain variability impacting our counts. We have added that detail into our methods (lines 221-222). If the reviewer suggests that we adjust the saturation of the images, we are happy to do that as well to make them look more consistent.

5 Figure 4. The cell complexity can be analyzed by FracLac, a convenient plugin of Fiji. It can measure the deformation of the cell outlines. Just a proposal.

 Thank you for bringing this option to our attention. We will look into FracLac for future studies.

6 Figure 7. Please provide the raw gel of wb as supplementary material.

We have now included the raw images for the MBP westerns used in Fig. 7 as a supplemental figure (Line 794-798).

Round 2

Reviewer 3 Report

All my concerns have been well addressed. It can be accepted now.